# Effect of Mycotoxin Cytochalasin A on Photosystem II in *Ageratina adenophora*

**DOI:** 10.3390/plants11202797

**Published:** 2022-10-21

**Authors:** Mengyun Jiang, Qian Yang, He Wang, Zhi Luo, Yanjing Guo, Jiale Shi, Xiaoxiong Wang, Sheng Qiang, Reto Jörg Strasser, Shiguo Chen

**Affiliations:** 1Weed Research Laboratory, Nanjing Agricultural University, Nanjing 210095, China; 2Bioenergetics Laboratory, University of Geneva, CH-1254 Geneva, Switzerland

**Keywords:** natural product, bioherbicide, photosynthetic inhibitor, chlorophyll *a* fluorescence, D1 protein, molecular docking

## Abstract

Biological herbicides have received much attention due to their abundant resources, low development cost, unique targets and environmental friendliness. This study reveals some interesting effects of mycotoxin cytochalasin A (CA) on photosystem II (PSII). Our results suggested that CA causes leaf lesions on *Ageratina adenophora* due to its multiple effects on PSII. At a half-inhibitory concentration of 58.5 μΜ (*I*_50_, 58.5 μΜ), the rate of O_2_ evolution of PSII was significantly inhibited by CA. This indicates that CA possesses excellent phytotoxicity and exhibits potential herbicidal activity. Based on the increase in the J-step of the chlorophyll fluorescence rise OJIP curve and the analysis of some JIP-test parameters, similar to the classical herbicide diuron, CA interrupted PSII electron transfer beyond Q_A_ at the acceptor side, leading to damage to the PSII antenna structure and inactivation of reaction centers. Molecular docking model of CA and D1 protein of *A. adenophora* further suggests that CA directly targets the Q_B_ site of D1 protein. The potential hydrogen bonds are formed between CA and residues D1-His215, D1-Ala263 and D1-Ser264, respectively. The binding of CA to residue D1-Ala263 is novel. Thus, CA is a new natural PSII inhibitor. These results clarify the mode of action of CA in photosynthesis, providing valuable information and potential implications for the design of novel bioherbicides.

## 1. Introduction

Weed control is always a great concern in agriculture due to its deduction in crop yields. In the last decades, chemical herbicides have been widely used to control the growth of weeds. However, some chemical herbicides have not been effective in controlling weeds due to the prevalence of resistant weeds because of long-term use [1]. In addition, chemical herbicides are polluting the environment and causing many negative effects on living organisms including human. These reasons have prompted attention to the development and application of novel bioherbicides which are more environmentally friendly. Natural compounds are extensively used in the development of bioherbicides. Several studies have shown that natural compounds derived from plants and microorganisms are effective in controlling weed seed germination and growth [2,3,4], becoming an important resource for the discovery of new natural herbicidal components and action targets.

Cytochalasins are a class of natural products produced by molds, including cytochalasin A, B, C, D, E, F, etc. More than one hundred cytochalasin-related structures or their derivatives have been identified and reported from molds like *Chaetomium*, *Rosellinia*, *Zygosporium*, *Helminthosporium*, *Penicillium*, *Aspergillus*, etc. [5]. In terms of structure, cytochalasins are typically characterized by a substituted perhydro-isoindolone moiety derived from a highly reduced polyketide backbone and amino acids, attached to a macrocyclic ring [5,6]. As for cytochalasin A (CA), the structure is shown in Figure 1A. It is worth mentioning that CA is an oxidized derivative of cytochalasin B (CB), and both differ only in the group attached at C-20 [7], which display as the carbonyl group and the hydroxyl group, respectively. Because of the structural similarities between cytochalasins, they have analogous biological activities. There are numerous studies on the effect of cytochalasins on animal cells. Cytochalasins are most well-known for targeting microfilaments [8] and thereby inhibiting various cellular processes, including actin polymerization [9,10,11], platelet agglutination [12], glucose transport [13,14], mitochondrial contraction [15], etc. In addition, the antitumor [16,17] activity of cytochalasins was also reported.

For the study of cytochalasins in plant cells, it has been reported that because of the action of cytochalasins on the actin cytoskeleton, they inhibit the motility of various organelles in plant cells, including chloroplasts [18,19], mitochondria [20], endoplasmic reticulum [21], Golgi stacks [22,23] and peroxisomes [24]. The disruption of the actin cytoskeleton by cytochalasins leads to activation of the salicylic acid pathway to enhance plant resistance to pathogens [25,26]. In addition, cytochalasin B and F were reported to be effective in inhibiting root elongation of wheat and tomato seedlings [27], showing an interesting herbicidal activity. Photosynthesis, one of the most critical physiological processes in green plants and the main target of many commercial herbicides, is also influenced by cellular cytochalasins. Several cytochalasins, including CA, have been shown to have the ability to inhibit photosynthesis. CA and CB reduced the light absorption capacity of the leaves, resulting in the suppression of photosynthetic efficiency [28]. Cytochalasin F inhibited the photosynthetic activity of Green Alga *Chlorella fusca* and was shown to be an inhibitor of photosynthesis [29]. Cytochalasin E was found to have a direct effect on photosynthesis by chlorophyll (Chl) fluorescence assay, probably due to the impairment of light harvesting [30]. In summary, we believe that cytochalasins are phytotoxic and have potential bioherbicidal activity. Thus, as a member of the cytochalasin family and being capable of inhibiting photosynthesis, CA’s mechanism of action in photosynthesis is highly inspiring for the development of innovative natural herbicides.

Although there are several reports on the effects of CA in plants, the specific mechanism of action of CA in plant cells is unclear, especially the target of photosynthesis inhibition. In photosynthetic reactions, photosystem II (PSII) is considered as the crucial photosynthetic component, catalyzing light-driven water-splitting and oxygen-evolving reaction, thereby converting light energy into electrochemical potential energy and generating molecular oxygen [31]. As is well known, many classical herbicides target PSII in competition with the native electron acceptor plastoquinone, binding at the Q_B_ site of the D1 protein, thereby blocking the transfer of electrons from Q_A_ to Q_B_ to inhibit photosynthesis [32,33]. Q_A_ and Q_B_ are the primary and secondary plastoquinone receptors in PSII, respectively. The highly conserved amino acid residues in the Q_B_ site may play a key role in the binding to PSII inhibitors in plants [34], and molecular docking studies have been reported in plants to identify the binding properties of some widely used commercial PSII inhibitory herbicides to the Q_B_ site within the D1 protein [34]. The molecular details of the interaction of many natural products possessing PSII inhibitory effects in the D1 Q_B_ site in plants remain to be elucidated. Therefore, it is important to explore the specific effects and sites of the natural compound CA on photosynthesis for the development of novel bioherbicides in the future.

In this work, we treated the leaves of *Ageratina adenophora* with CA and found that CA can cause leaf diseases. In addition, the oxygen evolution rate and photosynthetic activity of *Chlamydomonas reinhardtii* were reduced by CA. From these results, we inferred that CA has great potential for weed management due to its phytotoxicity and ability to inhibit PSII photosynthetic activity. Based on the study of Chl *a* fluorescence rise kinetics and the analysis of molecular docking, we aimed to determine the mode of action and the site of action of CA in PSII. It was found that CA interrupts PSII electron transfer beyond Q_A_ at the acceptor side, leading to impaired PSII antenna structure and inactivation of the reaction centers. The interaction model between the CA and D1 proteins of *A. adenophora* was simulated to discover the accurate binding site. The revelation of the effect of CA on PSII and the specific binding sites of CA in PSII probably helps to better utilize it as a tool for plant research, especially in the development of new effective photosynthetic herbicides.

## 2. Results and Discussion

### 2.1. CA Caused Leaf Lesions in A. adenophora and Reduced the Rate of Oxygen Evolution in C. reinhardtii

CA, a fungal phytotoxin, had an inhibitory effect on the growth and viability of poplar suspension-cultured cells because the dry weight and oxygen consumption of cells treated with CA were strongly inhibited [35]. To estimate the phytotoxicity of CA, the damage formation in mature leaves of *A. adenophora* after treatment with different concentrations of CA was monitored. The monitored lesion formation is shown in Figure 1B. At the highest concentration of 800 μM, the largest diameter of visible lesions on the leaves was observed, with necrosis and browning of the leaves in the treated area. The formation of necrosis or chlorosis implies severe damage to photosynthetic tissues from chlorophyll breakdown [36]. This suggested that CA possibly impaired the photosynthetic activity of *A. adenophora* leaves.

To verify the effect of CA on photosynthetic apparatus, we measured the PSII oxygen evolution rate of *C**. reinhardtii* cells incubated in different concentrations of CA. At a CA concentration of 100 μM, the rate of oxygen evolution was significantly reduced by approximately 64% (Figure 1C). The calculated *I*_50_ value (concentration required for 50% inhibition) was 58.5 μΜ. Diuron (3-(3′,4′-dichlorophenyl)-1,1-dimethylurea, DCMU), a classical PSII inhibitor, has been reported to completely inhibit the evolution of oxygen in *C. reinhardtii* cells at 1 μΜ [37] with an *I*_50_ value of 0.12 μM [38]. This suggests that CA has a good inhibitory effect on PSII, but is a weaker PSII inhibitor compared with diuron. In fact, diuron is a strong and specific PSII inhibitor, compared with which most natural photosynthetic inhibitors are less active [38]. It has been shown that the phytotoxin tenuazonic acid (TeA) and gliotoxin inhibit the rate of oxygen evolution with *I*_50_ values of 261 μM and 60 μM, respectively [36,39]. However, there are also stronger natural PSII inhibitors, such as sorgoleone, which have a similar ability to inhibit photosynthetic O_2_ evolution as diuron [40]. Thus, CA has an inhibitory effect on photosynthesis in plants and is a weaker PSII inhibitor.

### 2.2. CA Inhibited Photosynthesis Activity of C. reinhardtii

Chl *a* fluorescence is very sensitive to stress-induced changes in photosynthesis. To further demonstrate the inhibition of photosynthetic activity by CA, the changes in fluorescence parameters of CA-treated *C. reinhardtii* cells were monitored by Imaging-PAM fluorometer. Figure 2A shows the color-coded images of F_O_, F_M_ and F_V_/F_M_. F_O_, minimal fluorescence from dark-adapted leaf samples, indicates the level of fluorescence when Q_A_ is maximally oxidized, at which point the PSII centers are completely open [41]. Correspondingly, F_M_ indicates the level of fluorescence when Q_A_ is maximally reduced, at which point the PSII centers closed [41]. F_O_ usually represents phototropic pigment activity [42]; the decrease in F_M_ may be correlated with the quenching of fluorescence due to the oxidation of the plastoquinone pool or the damage of the PSII antennae [43]. F_V_/F_M_ represents the PSII maximum quantum yield. Under different concentration of CA treatments, images of F_O_ remained relatively stable, while images of F_M_ faded gradually. Under the treatment of CA, the F_V_/F_M_ image fades from dark blue to light blue. The values of the fluorescence parameter F_V_/F_M_ (Figure 2B) were slightly reduced but not significantly different. Electron transfer rate (ETR), a more sensitive [44] fluorescence parameter, showed significant concentration-dependent changes, with values decreasing with increasing CA concentration (Figure 2C). At the highest concentration of 40 μM CA, the ETR value was close to 0. The parameter values of ETR decreased approximately linearly, indicating that the inhibition of PSII electron transfer should be an important action site for CA on the photosynthetic apparatus. 

### 2.3. Fast Chl a Fluorescence Rise Kinetics OJIP of CA-Treated A. adenophora

For decades, Chl fluorescence rise kinetics has been widely used in studies of the structure and function of the photosynthetic apparatus because of its advantages of being fast, non-invasive and precise [41]. To further probe the accurate site of action of CA on PSII, we measured the fluorescence rise OJIP curves of CA-treated *A. adenophora* leaf discs (Figure 3). As shown in Figure 3A, the fluorescence rise curve of the control exhibited a typical polyphasic O-J-I-P shape. The fluorescence intensity increases from F_O_ (O-step) to F_M_ (P-step) and through two intermediate steps, F_J_ (J-step) and F_I_ (I-step). The initial fluorescence F_O_ at 20 μs, and the intermediate basic fluorescence data F_J_ and F_I_ at 2 and 30 ms, respectively. V_t_, representing relative variable fluorescence at time t, is calculated with the formula V_t_ = (F_t_ − F_O_)/(F_M_ − F_O_). The detailed parameters and definitions were listed in Table 1, which is according to Strasser et al. [41] and Chen et al. [45]. Under CA treatment, the shape of the fluorescence rise transient curve changed remarkably. As the concentration of CA and treatment time increased, the F_M_ decreased significantly and the I- and P-steps gradually disappeared (Figure 3A,C). The decrease of F_M_ is generally related to increased fluorescence quenching and the damage of antenna pigment [43]. Here, we proposed that CA affected the structure of antenna pigment so as to decrease the photosynthesis capacity.

To explore the information behind OJIP curves, the fluorescence data were double-normalized by F_O_ and F_M_, as relative variable fluorescence V_t_ = (F_t_ − F_O_)/(F_M_ − F_O_) (top) and ΔV_t_ = V_t(treatment)_ − V_t(control)_ (bottom) on logarithmic time scale (Figure 3B,D). Clearly, CA led to a concentration- and time-dependent rise of J-step, which is a classical phenomenon of the blockage of electron flow beyond the Q_A_ at the PSII acceptor side and the large accumulation of Q_A_^−^ [41]. Treatment of *A. adenophora* leaf discs with the specific PSII inhibitor diuron resulted in a rapid rise in J-step to the same F_M_ level at a concentration of 100 μM [38]. Apparently, the effect of CA on the kinetics of fluorescence rise is similar to that of diuron, mainly causing a significant increase in the J-peak. 

The elevated level of the J-step and the decrease in F_M_ suggest that CA not only blocked PSII electron flow beyond the Q_A_, but also impaired the structure and function of the PSII antenna.

### 2.4. The JIP-Test Analysis

The JIP-test based on the Theory of Energy Fluxes in Biomembranes is a powerful tool for quantifying the fast Chl fluorescence rise OJIP curve, providing access to the vitality of a photosynthetic sample [41]. To further elucidate the effect of CA on PSII, some functional parameters selected from the JIP-test were analyzed. Definitions and specific formulas for certain parameters are shown in Table 1. Obviously, the increase in CA concentration had little effect on the F_O_ but substantially reduced the F_M_ values. The ratio of F_V_/F_M_ showed a decline with the increase in CA treatment concentration (Figure 4A). This result is consistent with the results in Figure 2A,B. This implies that CA has a similar effect on the photosynthetic activity of *A. adenophora* and *C. reinhardtii*. We concluded that CA caused damage to the structure and function of the PSII antenna and thus inhibited plant photosynthetic activity. 

The value of V_J_ (variable fluorescence at J-step) and ΔV_J_ increased with increasing CA concentration (Figure 4B). At 200 μM, the V_J_ value increased to 0.79, which was more than twice as high as that of the control (0.36). The increase in V_J_ value indicates that CA blocks PSII electron transport beyond Q_A_. Additionally, the value of V_I_ (variable fluorescence at I-step) was 12% higher than that of the control at a CA concentration of 25 μM, but no obvious change in the value of V_I_ was observed as the CA concentration increased. 

In Figure 4C, a dramatic drop was observed in the parameters φ_Eo_ and ψ_Eo_, with a calculated *I*_50_ value of 83.25 μM and 103.86 μM for φ_Eo_ and ψ_Eo_, respectively. The φ_Eo_ denotes the quantum yield of PSII electron transfer, while ψ_Eo_ shows the probability that a trapped exciton moves an electron into the electron transport chain beyond Q_A_^−^ [41]. The decrease of φ_Eo_ and ψ_Eo_ values further supports that CA blocks the PSII electron transport beyond Q_A_. By increasing the CA treatment concentrations, the values of φ_Po_ also decreased, suggesting that the maximum quantum yield of PSII primary photochemistry was suppressed by CA. 

ABS/CS_0_ (absorption flux per excited leaf cross section), TR_0_/CS_0_ (light energy captured per unit leaf area) and ET_0_/CS_0_ (electron transport flux per leaf cross section) significantly declined under CA treatment [41] (Figure 4D). ABS/CS_0_ is available as a measure of average antenna size or chlorophyll concentration, and TR_0_/CS_0_ reflects the specific rate of exciton trapped per excited leaf cross-section by open reaction centers [46]. The decrease in ABS/CS_0_ and TR_0_/CS_0_ suggests that CA reduced the concentration of chlorophyll and disrupted the conformation of the antenna pigment assembly, thereby reducing the efficiency of light energy transfer between antenna pigment molecules and from PSII reaction centers. The quantum yield of electron transport is influenced, as shown by the decreases in ET_0_/CS_0_. The above results indicate that CA dramatically blocked the energy flux between photosynthetic structural units.

Since CA interrupts PSII electron flow from Q_A_ to Q_B_, decreases chlorophyll concentration and disrupts the conformation of the antenna pigment assembly, an inactivation event is expected to occur in the PSII reaction center. The fraction of Q_A_-reducing centers were calculated, Q_A_-reducing centers = [(RC/CS)_treatment_/(RC/CS)_control_] · [(ABS/CS)_treatment_/(ABS/CS)_control_]. In Figure 4E, the marked decrease in Q_A_-reducing centers suggests that CA did lead to a rapid closure of PSII reaction centers. A significant reduction in Q_A_-reducing centers certainly results in an increase in non-Q_A_-reducing centers, also known as heat sink centers or silent centers, which are radiators and are frequently used to protect the system from over-excitation and over-reduction, resulting in the generation of ROS [45]. The value of RC/CS_0_ (the density of Q_A_-reducing reaction centers per excited leaf cross section) also exhibited a similar decreased tendency after CA-treatment. This indicated that CA caused damage to the PSII reaction centers and inhibited the activity of PSII. 

It has been shown that the PSII reaction centers would become inactivated in the competitive substitution of the Q_B_ binding site in the D1 protein by PSII inhibitors, thus interrupting electron transport [41]. To investigate the binding of CA to the Q_B_ site of reaction centers, R_J_ was introduced to estimate the number of PSII reaction centers at the Q_B_ site populated by CA. The fluorescence parameter R_J_ could be derived from the equation R_J_ = [V_J(treatment)_ − V_J(control)_]/[1 − V_J(control)_] [45]. A visible concentration-dependent increase in R_J_ was observed after CA treatment (Figure 4E). The rise of R_J_ means that the increased CA molecules occupy the Q_B_ site of D1 protein, thus blocking the electron transfer from Q_A_ to Q_B_. Therefore, CA-induced inactivation of the PSII reaction center probably results from the binding of CA to D1 protein.

PI_ABS_, the performance index on absorption basis, is the most sensitive JIP-test parameter for the different stress treatments. Compared with mock, PI_ABS_ respectively decreased by 75% (25 μM), 86% (50 μM), 89% (100 μM) and 97% (200 μM) after 12 h of treatment with different concentrations of CA (Figure 4F). The other parameter DF_ABS_ showed a comparable downward trend. DF_ABS_ is derived from the equation DF_ABS_ = log(PI_ABS_) and can be defined as the total driving force of photosynthesis in the observed system [41]. These results reflected that the overall photosynthetic activity and efficiency of the leaves showed a concentration-dependent decrease after CA treatment.

Altogether, CA might destroy the structure of antenna pigment (F_O_, F_M_, ABS) and inhibit the energy flux between antenna units (ABS/CS_0_, TR_0_/CS_0_ and ET_0_/CS_0_) as well as electron transport above Q_A_ (φ_Eo_ and ψ_Eo_) by binding to the Q_B_ binding site of D1 protein (R_J_) and thus lead to the dramatic decrease of overall photosynthesis activity (PI_ABS_) in *A. adenophora*. To verify the primary site of action of CA on PSII, the correlations between some selected parameters are shown in Figure 5. As shown in Figure 5A, the data displayed a highly linear relationship between F_M_ and ABS/CS_0_ or TR_0_/CS_0_ in the leaves treated with different concentrations of CA. It suggested that the CA-caused decrease of F_M_ was induced by the concentration-dependent decrease of ABS/CS_0_ or TR_0_/CS_0_. So, the inhibition of CA on the efficiency of light energy transfer between antenna pigment molecules and from PSII reaction centers was the main reason for the decrease in F_M_. V_J_ showed a high negative correlation with φ_Eo_ and a high positive correlation with R_J_ (Figure 5B), which suggested that CA occupied the Q_B_ sites in D1 protein to block the PSII electron transport beyond Q_A_, thus leading to a large accumulation of Q_A_^−^ as an increase of V_J_. In addition, the linear relationship between the parameters R_J_ and ET_0_/CS_0_, φ_Eo_ and ψ_Eo_ further demonstrated that CA interferes with electron transport by occupying the Q_B_-binding site (Figure 5C). Under CA treatment, the decrease of the parameter Q_A_-reducing centers was correlated with ET_0_/CS_0_ and ψ_Eo_, respectively (Figure 5D). It indicated that CA caused the interruption of PSII electron transport beyond the Q_A_, which resulted in the inactivation of the reaction centers.

To estimate the structural and functional contribution of PSII to PI_ABS_, the relationships between PI_ABS_ and ET_0_/CS_0_, TR_0_/CS_0_ and ABS/CS_0_ are presented (Figure 5E). As a result, CA-caused PI_ABS_ decrease was highly related to the reduction of ABS/CS_0_and ET_0_/CS_0_, especially for ET_0_/CS_0_, but not TR_0_/CS_0_. In other words, CA-caused loss of PSII overall photosynthetic activity is mainly due to the decrease of PSII electron transport efficiency, and secondly attributed to chlorophyll concentration or the disruption of the antenna pigment assembly. This is strongly supported by the high linear correlation between PI_ABS_ and ψ_Eo_ or Q_A_-reducing centers in the presence of different concentrations of CA. However, there is not an evident linear relationship between φ_Po_ and PI_ABS_ (Figure 5F). Thus, the significant decrease of overall photosynthetic activity of PSII is really a result of antenna destruction, electron transfer interruption, and inactivation of PSII Q_A_-reducing centers in the presence of CA. 

In conclusion, CA acts in a way similar to the classical diuron-like herbicides, targeting mainly the Q_B_ site of the D1 protein and blocking PSII electron transport at the acceptor side to inhibit photosynthetic efficiency in plants.

### 2.5. Modeling of CA Binding to D1 Protein

In oxygenated photosynthesis in plants, algae and cyanobacteria, PSII undergoes light-driven plastoquinone reduction by electrons from water. Nevertheless, PSII herbicides interrupt this process. Numerous previous studies have shown that herbicides targeting PSII inhibit electron transfer from Q_A_ to Q_B_ by competing with native plastoquinone for its Q_B_ binding niche in the D1 protein [32,47]. The D1 protein in higher plants contains mainly five transmembrane α-helices, and the Q_B_ binding site is located between the fourth and fifth transmembrane helices [32,38,48]. The amino acids that form the Q_B_ and herbicide-binding niche of D1 protein are from Phe211 to Leu275 [33]. 

To further verify whether CA binds to the Q_B_ site of D1 protein, CA was modeled to the Q_B_ site of *A. adenophora* D1 protein based on experimental and theoretical structural information provided by Discovery Studio (version 2016, BIOVIA, San Diego, CA, USA) [49] on the binding of D1 protein by PSII herbicides (Figure 6). The molecular model of CA binding to D1 protein showed that hydrogen bonds were formed between the ethylamino hydrogen (NH) of the residue D1-His215 and the O3 oxygen atom of CA, as well as between the C18-OH hydroxyl oxygen atom of CA and CO carbonyl oxygen atom of the residue D1-Ala263 or CO carbonyl oxygen atom of D1-Ser264, respectively. Their bonding distance was 2.48 Å, 2.44 Å and 2.35 Å, respectively (Figure 6A and Table 2). The residues D1-Phe211, D1-Phe265 and D1-Phe274 also formed, respectively, a Pi hydrophobic interaction with the C16, C17=CH2 and the C16 of CA with the bound distance of 3.97 Å, 3.83 Å and 3.86 Å (Table 2). Additionally, the alkyl hydrophobic interactions between CA and the residues D1-Ile248, D1-Leu251, D1-His252 and D1-Leu271 were also probably involved in the complex stabilization of CA binding to the Q_B_ site (Table 2). Figure 6B illustrates the docked pose of CA at the Q_B_ binding site of *A. adenophora* D1 protein. Hydrogen bonding with D1-Ser264 is a favorable orientation for the classical PSII herbicides atrazine and diuron [50]. Apparently, CA is not exactly the same as atrazine and diuron in the binding environment, although all target the Q_B_ site of the D1 protein. 

It is well known that PSII inhibitor herbicides share the same action target D1 protein, but each herbicide has its own specific orientation in binding to the Q_B_ niche. The classic inhibitors of PSII can be classified as the urea/triazine, and phenolic inhibitors. Extensive studies with herbicide-tolerant mutants and crystal structure-based molecular modeling have shown that hydrogen bonding with D1-Ser264 is the key interaction for urea/triazine herbicides to bind D1 proteins, whereas the favorable orientation of phenolic herbicides to bind D1 proteins is through hydrogen bonding with His215 [32,48,51]. Here, D1-His215, D1-Ser264 and D1-Ala263 are thought to be involved in the binding pocket of CA and provide hydrogen bonds for CA (Figure 6B). As compared with the two types of PSII herbicides, the binding environment of CA in D1 protein overlaps but differs from theirs, and residue D1-Ala263 is a novel site of action. Overall, the binding orientation of CA to D1 protein appears to be more complex than that of classical herbicides to D1 protein.

Some natural products with herbicidal properties also exhibit complex binding orientations to the Q_B_ site. According to the constructed molecular model of patulin (PAT) [38] binding to D1 protein, D1-His252 provides a major hydrogen bond to the O_2_ carbon-based oxygen atom of PAT. This mode of binding to D1 proteins is not consistent with either urea/triazine family inhibitors or phenolic inhibitors and may be the result of differences in the characteristic groups. The molecular interaction model of TeA [50] and *Arabidopsis* D1 protein indicates that the residue Gly256 of D1 protein plays a key role in TeA binding to the Q_B_ niche of D1 protein. Apparently, this binding behavior is also different from that of classical herbicides. In this study, the molecular interaction model of CA binding to the Q_B_ site indicated that residues D1-His215, D1-Ala263 and D1-Ser264 were involved in the formation of hydrogen bonds, respectively, promoting the binding of CA to D1 protein. Compared with the classical PSII herbicides and natural PSII inhibitors mentioned above, CA has a unique binding orientation for residue D1-Ala263 at the Q_B_ site. Therefore, CA is a novel natural photosynthetic inhibitor with potential for future development as a bioherbicide. Further verification of the exact binding environment of CA, however, is required by crystallographic data and mutant experiments.

## 3. Materials and Methods

### 3.1. Plants and Chemicals

Invasive weed *A. adenophora* was germinated on a mixture of perlite-vermiculite−peat (0.5:1:3, *v*/*v*) at 20–25 °C under 200 μmol (photons) m^−2^ s^−1^ white light (day/night, 12 h/12 h) and 70% relative humidity in the greenhouse. After 180 days of growth, the leaves of plants were sampled. 

*C. reinhardtii* wild-type strain was obtained from the Freshwater Algae Culture Collection at the Institute of Hydrobiology (FACHB-col-lection, Chinese Academy of Science, China). Cells were grown in liquid Tris-acetate-phosphate (TAP) medium at 25 °C under white light (100 μmol (photons) m^−2^ s^−1^) with a 12 h photoperiod and shaken once every 12 h. The total biomass was determined by optical density of cell suspensions in a spectrophotometer (UV-1800, Shimadzu, Kyoto, Japan) at 750 nm (A_750_). The cells of the logarithmic phase (approximately 3 or 4 days, A_750_ of about 0.65) were collected and washed twice with distilled water, then resuspended with buffer A. The buffer contained 20 mM HEPES-KOH (pH 7.5), 350 mM sucrose and 2.0 mM MgCl_2_. The cells were used for the subsequent experiments. 

CA (CAS No. 14110-64-6) was purchased from Sigma-Aldrich (Shanghai, China), and other common chemical reagents used in this work were purchased from Aladdin (Shanghai, China). CA was prepared in 100% acetone and diluted in distilled water as required. The final concentration of acetone in each experiment did not exceed 10% (*v/v*).

### 3.2. Phytotoxicity Assessment 

For the normal images of *A. adenophora*, intact detached-leaves from healthy plants of *A. adenophora* were washed with distilled water, dried with sterilized-filter papers, and transferred onto wet sterilized-filter papers in Petri dishes. Leaves were slightly punctured on their abaxial margin with a needle. The 20 μL droplet of 0 (mock, 10% acetone), 100, 200, 400 and 800 μM CA solutions were added onto the wound site of leaves. All samples were maintained at 25 °C for 12 h in a growth chamber, under approximate 200 μmol (photons) m^−2^ s^−1^ white light (day/night, 12 h/12 h). Results were recorded with a Canon G15 camera (Canon, Tokyo, Japan) and diameters of leaf necrotic lesions were determined with a vernier caliper (ROHS HORM 2002/95/EC, Xifeng, Qingyang, China).

### 3.3. Measurement of PSII Oxygen-Evolution Rate of C. reinhardtii

A Clark-type oxygen electrode (Hansatech Instruments Ltd., King’s Lynn, UK) was used to determine the oxygen evolution rate of *C. reinhardtii* according to [39]. Before measurements, CA was added to 2 mL cell suspensions (A_750_ of 0.65) to make final concentrations as indicated (0, 12.5, 25, 50 and 100 μM), and then the cells were incubated for 3 h in darkness at 25 °C. Treated cells containing 45 μg chlorophylls were added into the PSII reaction medium of 4.0 mL, which included 50 mM HEPES-KOH buffer (pH 7.6), 4 mM K_3_Fe(CN)_6_, 5 mM NH_4_Cl and 1 mM p-phenylenediamine. After the reaction mixture was illuminated with 400 μmol m^−2^ s^−1^ red light for 1 min, the oxygen evolution rate was recorded. The independent experiment was repeated three times. 

### 3.4. Chl a Fluorescence Imaging

A MAXI-version of the pulse-modulated Imaging-PAM M-series fluorometer (Heinz Walz GmbH, Effeltrich, Germany) was used to determine the Chl fluorescence imaging [44]. Before measurements, samples were adapted for 0.5 h in the dark under the imaging system camera after focusing the camera. Under 0.25 μmol (photons) m^−2^ s^−1^ measuring light, 110 μmol (photons) m^−2^ s^−1^ actinic light and 6000 μmol (photons) m^−2^ s^−1^ saturation pulse light, fluorescence images were monitored. The following parameters were measured directly: F_V_/F_M_ and ETR.

For the measurement of *C. reinhardtii* cells, 200 μL of suspensions (A_750_ of about 0.65) with 10% acetone (mock) and CA were added into the 96-well black microtiter plate to a final CA concentration of 20, 30 and 40 μM, respectively. The treated cells were incubated for 2.5 h at 25 °C under 100 μmol (photons) m^−2^ s^−1^ white light, and then samples were adapted in darkness for 0.5 h before fluorescence determination.

### 3.5. Chl a Fluorescence Kinetics OJIP Curves and JIP-Test

A plant efficiency analyzer (Handy PEA, Hansatech Instruments Ltd., King’s Lynn, Norfolk, UK) was used to measure the Chl *a* fluorescence kinetics OJIP curves. For the measurement, 7-mm diameter leaf discs of *A. adenophora* were washed with distilled water and incubated for 12 h in CA solutions with 0 (10% acetone, mock), 25, 50, 100 and 200 μM concentrations, respectively. Meanwhile, to verify the time-dependent effect of CA, leaf discs were also treated with 50 μM CA for 0, 3, 6 and 12 h, respectively. Samples were well adapted to darkness for 0.5 h prior to measurement. Raw fluorescence transients OJIP was transferred to a spreadsheet by Handy PEA V1.30 and BiolyzerHP3 software. The independent experiment was repeated three times with around 15 repetitions. The raw data of the fluorescence kinetics OJIP curves were analyzed using the JIP-test [41,52]. Detailed JIP-test parameters containing formulas, equations and definitions are listed in Table 1, which are according to Strasser et al. [41] and Chen et al. [45].

### 3.6. Modeling of CA in D1 Protein of A. adenophora

The target D1 protein of *A. adenophora* can be provided on the basis of the amino acid sequence (reference sequence no. YP 004564352.1). The sequence of amino acid was obtained from NCBI, in FASTA format. Evolutionary related protein structures were searched for by the BLAST databases through the SWISS-MODEL Template Library (SMTL) [53]. Searched templates of D1 protein were estimated using Global Model Quality Estimate (GMQE) and Quaternary Structure Quality Estimate (QSQE) and ranked by the expected quality of the resulting models. The protein structures of top-ranked templates were selected from the Protein Data Bank (PDB) and built the homology model of *A. adenophora* D1 protein by the Protein Module of Discovery Studio. The chemical structure of CA was constructed with the software of ChemBioDraw Ultra 14.0 (CambridgeSoft, Cambridge, MA, USA), and the Chem3D Pro 14.0 (CambridgeSoft, Cambridge, MA, USA) was used to minimize energy. The docking was performed by DS-CDocker in Discovery Studio 2016 (BIOVIA, San Diego, CA, USA). The polar was added to the protein during energy minimization and molecular refinement.

### 3.7. Statistical Analysis

One-way ANOVA was carried out and means were separated by DUNCAN LSD at 95% using SPSS Statistics 20.0.

## 4. Conclusions

CA is a natural mycobacterial metabolite that is widely known for its targeting of actin microfilaments to produce a series of interesting physiological phenomena. Our study reveals some interesting effects of CA on plant PSII and finds that CA has potential herbicidal activity. Based on the analysis of Chl *a* fluorescence rise kinetics and PSII oxygen evolution rate, CA inhibits photosynthesis by occupying the Q_B_ site in the D1 protein, thus blocking the electron transfer from Q_A_ to Q_B_. The loss of overall photosynthetic activity of PSII is mainly due to the reduction of electron transfer efficiency, the destruction of antenna pigments and the inactivation of PSII reaction centers in the presence of CA. This mode of action is similar to that of the classical PSII herbicide diuron. The molecular docking model of CA binding to *A. adenophora* D1 protein further suggests that the direct target of CA is the Q_B_ site of D1 protein in PSII. It is proposed that CA forms potential hydrogen bonds with residues D1-His215, D1-Ala263 and D1-Ser264, respectively. Among them, D1-Ala263 is a novel site of action of CA different from the urea/triazine and the phenolic inhibitors. The inhibitory effect of CA on PSII allows CA to be used directly as a PSII inhibitor in various studies in the future. Considering the high development costs of chemical herbicides and the prevalence of resistant weeds, the unique binding of CA to D1 protein provides a new idea for the design of novel and efficient herbicide molecules to effectively manage weeds.

## Figures and Tables

**Figure 1 plants-11-02797-f001:**
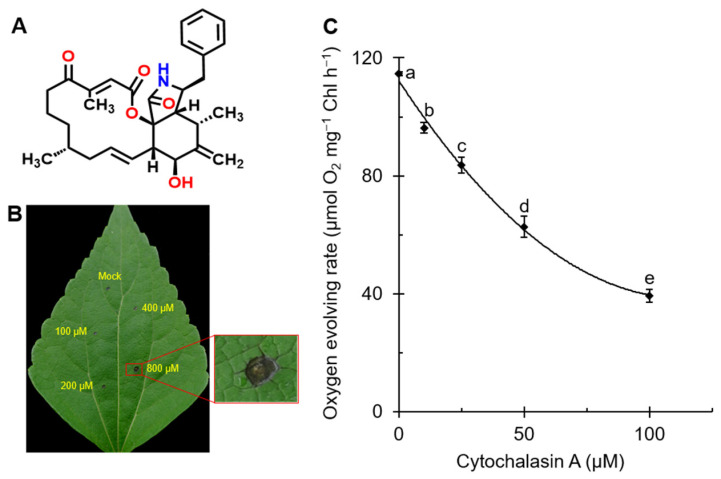
(**A**) Chemical structure of cytochalasin A (C_29_H_35_NO_5_, MW = 477.6). (**B**) Lesion formation of *A. adenophora* leaves treated with mock (10% acetone) and 100, 200, 400 and 800 μM cytochalasin A. (**C**) Effect of cytochalasin A on oxygen evolving rate of *C. reinhardtii* cells. Data shown are mean values ± SE of three independent measurements with around 15 repetitions. The different lowercase letters (a, b, c, d, e) indicate to be statistical significance at 0.05 level.

**Figure 2 plants-11-02797-f002:**
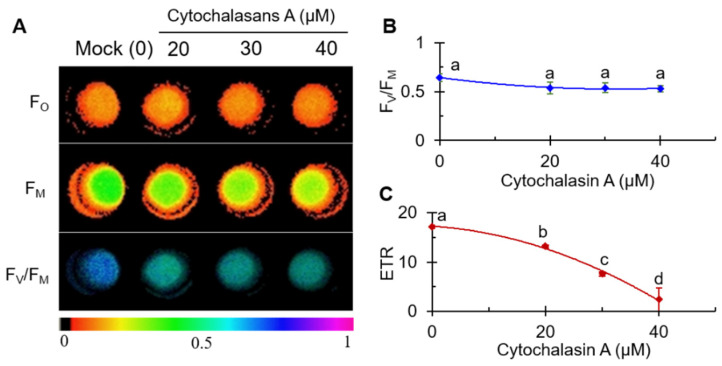
Effects of cytochalasin A on color fluorescence imaging of F_O_, F_M_ and F_V_/F_M_ (**A**), the value of the maximum quantum yield of PSII (F_V_/F_M_) (**B**), and electron transport rate (ETR) (**C**) of *C. reinhardtii* cells. Fluorescence images were indicated by color code in the order of black (0) through red, orange, yellow, green, blue, violet to purple (1). The number codes above images are marked from 0 to 1, showing the changes. Each value is the average ± SE of three independent experiments with around 15 repetitions. The different lowercase letters (a, b, c, d) indicate to be statistical significance at 0.05 level.

**Figure 3 plants-11-02797-f003:**
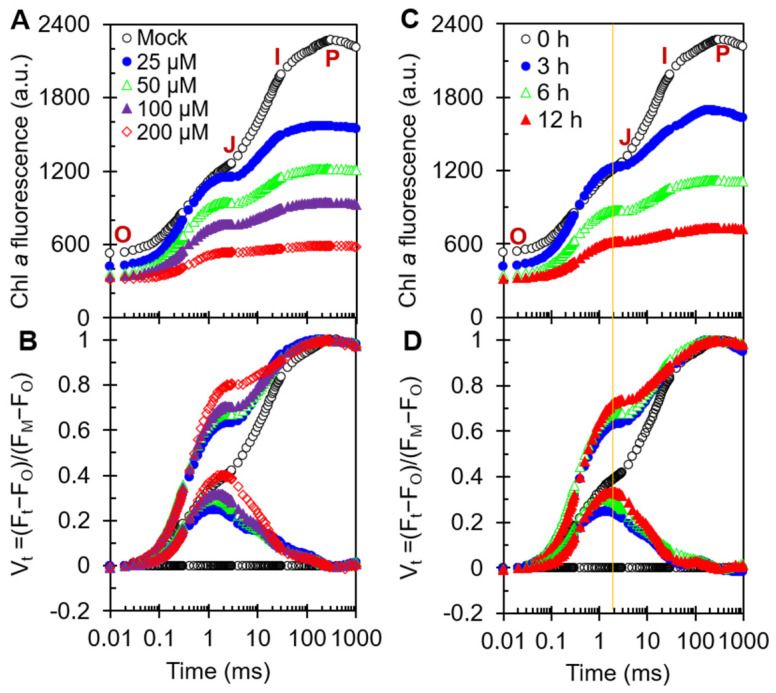
Effects of cytochalasin A on fast chlorophyll *a* fluorescence rise kinetics OJIP of *A. adenophora* leaf discs. (**A**,**B**) Raw Chl *a* fluorescence rise kinetics and its variable changes normalized by F_O_ and F_M_ as V_t_ = (F_t_ − F_O_)/(F_M_ − F_O_) (top) and ∆V_t_ = V_t(treatment)_ − V_t(control)_ (bottom) in a logarithmic time scale of *A. adenophora* treated with different concentrations (0, 25, 50, 100 and 200 μM) of cytochalasin A for 12 h. (**C**,**D**) Raw Chl *a* fluorescence rise kinetics and its variable changes normalized by F_O_ and F_M_ as V_t_ = (F_t_ − F_O_)/(F_M_ − F_O_) (top) and ∆V_t_ = V_t(treatment)_ − V_t(control)_ (bottom) in a logarithmic time scale of *A. adenophora* treated with 50 μM cytochalasin A for different time periods (0, 3, 6 and 12 h). Each curve is the average of three independent experiments with around 15 repetitions.

**Figure 4 plants-11-02797-f004:**
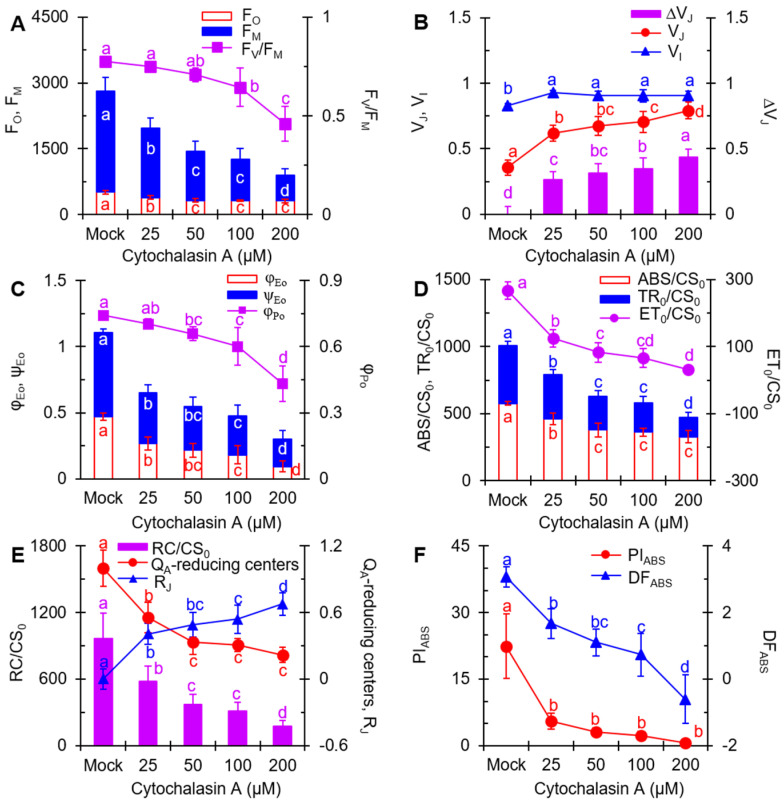
Changes of selected JIP-test fluorescence parameters of *A. adenophora* leaf discs treated with different concentrations of cytochalasin A. (**A**) F_O_, F_M_ and F_V_/F_M_. (**B**) V_J_, V_I_ and ∆V_J_. (**C**) φ_Eo_, ψ_Eo_ and φ_Po_. (**D**) ABS/CS_0_, TR_0_/CS_0_ and ET_0_/CS_0_. (**E**) RC/CS_0_, Q_A_-reducing centers, R_J_. (**F**) PI_ABS_ and D.F. Each curve is the average of three independent experiments with around 15 repetitions. The different lowercase letters (a, b, c, d) indicate to be statistical significance at 0.05 level.

**Figure 5 plants-11-02797-f005:**
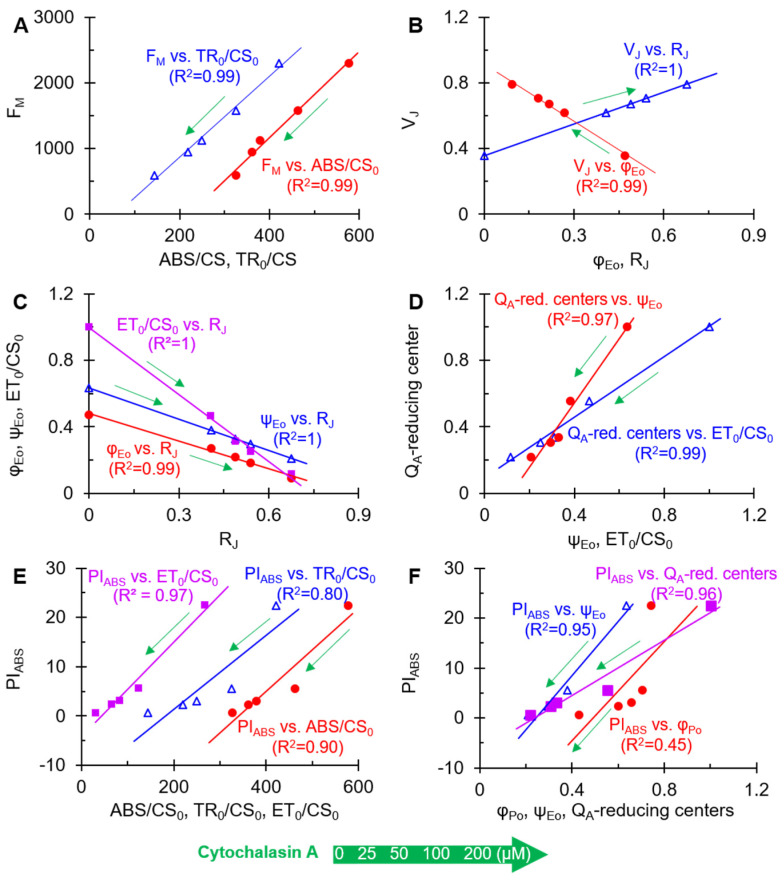
Analysis of the linear relationship between different selected JIP-test parameters after *A. adenophora* leaf discs were treated with different concentrations of cytochalasin A for 12 h. (**A**) The ABS/CS and TR_0_/CS versus F_M_. (**B**) The φ_Eo_ and R_J_ versus V_J_. (**C**) The R_J_ versus φ_Eo_, ψ_Eo_ and ET_0_/CS_0_. (**D**) The ψ_Eo_ and ET_0_/CS_0_ versus Q_A_-reducing centers. (**E**) The ABS/CS_0_, TR_0_/CS_0_ and ET_0_/CS_0_ versus PI_ABS_ (**F**) The φ_Po_, ψ_Eo_ and Q_A_-reducing centers versus PI_ABS_. Each curve is the average of three independent experiments with around 15 repetitions.

**Figure 6 plants-11-02797-f006:**
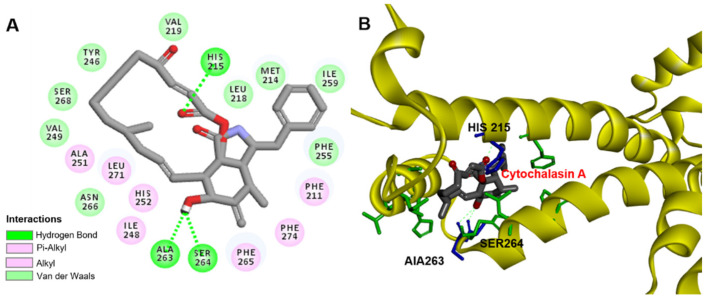
The simulated modeling of cytochalasin A binding to D1 protein of *A. adenophora*. (**A**) Hydrogen bonding interactions for cytochalasin A binding to D1 protein. (**B**) Stereo view of cytochalasin A binding environment of D1 protein. Here, carbon atoms are shown in grey, nitrogen atoms in blue, oxygen in red and hydrogen atoms in white. The possible hydrogen bonds are indicated by dashed lines.

**Table 1 plants-11-02797-t001:** Formulae and explanation of the technical data of the OJIP curves and the selected JIP-test parameters used in this study ^1^.

Technical Fluorescence Parameters	
F_t_	fluorescence at time t after onset of actinic illumination
F_O_ ≅ F_20μs_	minimal fluorescence, when all PSII RCs are open
F_L_ ≡ F_150μs_	fluorescence intensity at the L-step (150 μs) of OJIP
F_K_ ≡ F_300μs_	fluorescence intensity at the K-step (300 μs) of OJIP
F_J_ ≡ F_2ms_	fluorescence intensity at the J-step (2 ms) of OJIP
F_I_ ≡ F_30ms_	fluorescence intensity at the I-step (30 ms) of OJIP
F_P_ (= _M_)	maximal fluorescence, at the peak P of OJIP
F_v_ ≡ F_t_ − F_O_	variable fluorescence at time t
V_t_ ≡ (F_t_ − F_O_)/(F_M_ − F_O_)	relative variable fluorescence at time t
V_J_ = (F_J_ − F_O_)/(F_M_ − F_O_)	relative variable fluorescence at the J-step
V_I_ = (F_I_ − F_O_)/(F_M_ − F_O_)	relative variable fluorescence at the I-step
**Quantum efficiencies or flux ratios**	
φ_Po_ = PHI(P_0_) = TR_0_/ABS = 1 − F_O_/F_M_	maximum quantum yield for primary photochemistry
ψ_Εo_ = PSI_0_ = ET_0_/TR_0_ = (1 − V_J_)	probability that an electron moves further than Q_A_^−^
φ_Eo_ = PHI(E_0_) = ET_0_/ABS = (1 − F_O_/F_M_) (1 − V_J_)	quantum yield for electron transport (ET)
**Phenomenological energy fluxes (per excited leaf cross-section-CS)**	
ABS/CS = Chl/CS	absorption flux per CS
TR_0_/CS = φ_Po_·(ABS/CS)	trapped energy flux per CS
ET_0_/CS = φ_Po_·ψ_Εo_·(ABS/CS)	electron transport flux per CS
**Density of reaction center (Q_A_-reducing PSII reaction center–RC)**	
RC/CS = φ_Po_·(V_J_/M_0_)·(ABS/CS)	density of Q_A_-reducing PSII RCs per CS
Q_A_-reducing centers = (RC/RC_reference_)·(ABS/ABS_reference_) = [(RC/CS)_treatment_/(RC/CS)_control_]·[(ABS/CS)_treatment_/(ABS/CS)_control_]	fraction of Q_A_-reducing PSII RCs
R_J_ = (ψ_Εo(control)_ − ψ_Εo(treatment)_)/ψ_Εo(control)_ = (V_J(treatment)_ − V_J(control)_)/(1 − V_J(control)_)	number of PSII RCs with Q_B_-site filled by PSII inhibitor
**Performance indexes**	
$PI_{ABS}$ ≡ $\frac{\gamma_{\mathrm{RC}}}{1-\gamma_{\mathrm{RC}}}\cdot \frac{\varphi_{\mathrm{Po}}}{1-\varphi_{\mathrm{Po}}}\cdot \frac{\Psi_{\mathrm{Eo}}}{1-\Psi_{\mathrm{Eo}}}$	performance index for energy conservation from photons absorbed by PSII to the reduction of intersystem electron acceptors
**Phenomenological energy fluxes (per excited leaf cross-section-CS)**	
${\mathrm{DF}}_{\mathrm{ABS}}=\log\left({\mathrm{PI}}_{\mathrm{ABS}}\right)=\log\left(\frac{\mathrm{RC}}{\mathrm{ABS}}\right)+\log(\frac{\phi_{\mathrm{Po}}}{1-\phi_{\mathrm{Po}}})+\log(\frac{\psi_{\mathrm{Eo}}}{1-\psi_{\mathrm{Eo}}})$	driving force on absorption basis

^1^ Subscript “0” (or “o” when written after another subscript) indicates that the parameter refers to the onset of illumination, when all RCs are assumed to be open.

**Table 2 plants-11-02797-t002:** Possible bonding interactions for CA binding to the D1 protein of *A. adenophora*.

CA Chemical Structure	Donor	Acceptor	Interactions	Bound Distance (Å)
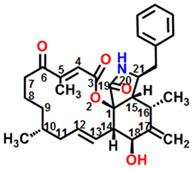	D1-Phe 211	CAC16-CH_3_	Pi Hydrophobic	3.97
	D1-His 215 NH	CA O3	Hydrogen Bond	2.48
	D1-Ile 248	CA C10-CH_3_	Alkyl Hydrophobic	3.72
	D1-Leu 251	CA C11	Alkyl Hydrophobic	3.77
	D1-His 252	CA C10-CH_3_	Alkyl Hydrophobic	3.81
	CA C18-OH	D1-Ala 263 CO	Hydrogen Bond	2.44
	CA C18-OH	D1-Ser 264 CO	Hydrogen Bond	2.35
	D1-Phe 265	CA C17=CH_2_	Pi Hydrophobic	3.83
	D1-Leu 271	CA C11	Alkyl Hydrophobic	3.91
	D1-Phe 274	CA C16-CH_3_	Pi Hydrophobic	3.86

## Data Availability

Not applicable.

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
