# Peer review of "Effect of Mycotoxin Cytochalasin A on Photosystem II in *Ageratina adenophora"

_plants, 2022, doi:10.3390/plants11202797_

Round 1
Reviewer 1 Report
Dear authors,
This study is interesting to the audience, particularly for the new role of mycotoxin cytochalasin A (CA) for plants. This study may contribute to the complex networks of bioherbicides.
Line 84-92: I suggest to rewrite this paragraph, because this statement is more like for conclusion, not in the introduction. This paragraph should indicate your hypothesis and research goals.
Author Response
Thank you for your suggestion. We have rewritten this part based on your suggestion. (Line 94-105)
Reviewer 2 Report
The paper manuscript “Effect of mycotoxin cytochalasin A on photosystem II in Ager- 2
atina adenophora.”
Overall, this is a well written manuscript and has a potential to be accepted. Nevertheless, the authors should revise and expand their Conclusion about the mode of action of the mycotoxin cytochalasin (CA).
Minor comments follow.
1)General Comment: Please check abbreviations with consistency in main text. Define it at the first appearance, then use it after the definition (QA, FO, FM, FV/FM, etc.).
2)Please enrich the introduction with more recent references, if available.
3) Line 43-44: Please rephrase the sentence.
4) Line 49, 340: See General comment above.
5) Line 79: See General comment above.
6) Line 169: See General comment above.
7) Lines 274-275 : Remove the empty line.
8) Line 324: “……..provided by the Discovery Studio version…”Please provide additional reference(s), if available..
9) Page 377: Conclusion: It is proposed to be removed in the end of the manuscript after Materials and methods (under point 4).
I will be glad to provide further details if needed and thank you for contacting me.
